# Identification of Optimal Reference Genes for qRT-PCR Normalization for Physical Activity Intervention and Omega-3 Fatty Acids Supplementation in Humans

**DOI:** 10.3390/ijms24076734

**Published:** 2023-04-04

**Authors:** Agata Grzybkowska, Katarzyna Anczykowska, Jędrzej Antosiewicz, Szczepan Olszewski, Magdalena Dzitkowska-Zabielska, Maja Tomczyk

**Affiliations:** 1Faculty of Physical Education, Gdansk University of Physical Education and Sport, 80-336 Gdansk, Poland; agata.grzybkowska@awf.gda.pl (A.G.); katarzyna.anczykowska@awf.gda.pl (K.A.); magdalena.dzitkowska-zabielska@awf.gda.pl (M.D.-Z.); 2Department of Bioenergetics and Physiology of Exercise, Medical University of Gdansk, 80-211 Gdansk, Poland; jant@gumed.edu.pl (J.A.); szczepan1olszewski@gumed.edu.pl (S.O.); 3Center of Translational Medicine, Medical University of Gdansk, 80-952 Gdansk, Poland

**Keywords:** gene expression, mRNA levels, polyunsaturated fatty acids, n-3 PUFAs, endurance training

## Abstract

The quantitative polymerase chain reaction (qRT-PCR) technique gives promising opportunities to detect and quantify RNA targets and is commonly used in many research fields. This study aimed to identify suitable reference genes for physical exercise and omega-3 fatty acids supplementation intervention. Forty healthy, physically active men were exposed to a 12-week eicosapentaenoic acid (EPA) and docosahexaenoic acid (DHA) supplementation and standardized endurance training protocol. Blood samples were collected before and after the intervention and mRNA levels of six potential reference genes were tested in the leukocytes of 18 eligible participants using the qRT-PCR method: GAPDH (Glyceraldehyde-3-phosphate dehydrogenase), ACTB (*Beta actin*), TUBB (*Tubulin Beta Class I*), RPS18 (*Ribosomal Protein S18*), UBE2D2 (*Ubiquitin-conjugating enzyme E2 D2*), and HPRT1 (*Hypoxanthine Phosphoribosyltransferase 1*). The raw quantification cycle (Cq) values were then analyzed using RefFinder, an online tool that incorporates four different algorithms: NormFinder, geNorm, BestKeeper, and the comparative delta-Ct method. Delta-Ct, NormFinder, BestKeeper, and RefFinder comprehensive ranking have found GAPDH to be the most stably expressed gene. geNorm has identified TUBB and HPRT as the most stable genes. All algorithms have found ACTB to be the least stably expressed gene. A combination of the three most stably expressed genes, namely GAPDH, TUBB, and HPRT, is suggested for obtaining the most reliable results.

## 1. Introduction

Quantitative polymerase chain reaction (qRT-PCR) is a versatile technique that is widely used in many research fields to determine the relative change in mRNA levels of tested genes. Gene expression analysis has become more affordable and accessible over the last 15 years [1]. Evaluating gene expression in human leukocytes has been previously used in, e.g., cancer [2], multiple sclerosis research in human cells [3], as well as in other species [4]. However, its application in studies involving physical training and supplementation is limited. A salient feature of qRT-PCR is the determination of relative gene expression results represented as a quantification cycle (Cq) value. The Cq value indicates the PCR cycle number at which the fluorescent signal generated by the amplification of the target gene surpasses the background fluorescence level. Obtaining relative results means that data need to be normalized with at least two or three stably expressed genes, called reference genes. However, the Cq value is not the only result obtained from qRT-PCR and should be taken into consideration together with other key measurements, such as amplification efficiency and melting curve analysis.

The identification of stably expressed genes in human cells, that could be used as reference genes in research, is fundamental for obtaining reliable and reproducible results, yet the use of a single reference gene without proof of validation is common in literature [5].

It has also been demonstrated that reference genes must vary for different types of cells and interventions, as no single gene could be used as a reference [6]. Hence, the use of multiple (usually two or three) reference genes should be adopted as a gold standard, as it significantly reduces the risk of producing artefactual data [7,8]. MIQE (Minimum Information for Publication of Quantitative Real-Time PCR Experiments) guidelines state that reference genes should be chosen and validated according to the specific experimental design [9]. Using several mathematic algorithms is in line with this recommendation. Physical activity can influence leukocytes, as they are powerful mediators that produce chemokines, cytokines, and growth factors, which are crucial for recovery and adaptive processes [10]. Similarly, omega-3 polyunsaturated fatty acids (n-3 PUFAs) are proven to influence DNA (Deoxyribonucleic acid) methylation changes in leukocytes, which is one way of regulating gene expression [11]. These two factors are considered to be promising targets for improving human health and could influence a great number of mRNA levels in leukocytes. There is growing evidence that increasing the amount of omega-3 fatty acids in the diet, particularly EPA and DHA acids, can promote a number of health benefits, including but not limited to the regulation of vascular and immune functions or inflammation [12]. However, there are few long-term studies in physically active individuals evaluating the effects of omega-3 fatty acids supplementation along with endurance training, on physical performance indicators [13], which was the aim of our previous work [14]. The pleiotropic effect of omega-3 fatty acids in the human body has been confirmed by published papers [15]. It has also been proven on a molecular level. In this regard, Bouwens et al., found that high doses of EPA and DHA supplementation altered the expression of 1040 genes [16]. This indicates the complexity and importance of finding genes that are not affected by omega-3 ingestion and thus, present stable expression in qRT-PCR research.

The purpose of the present study was to identify stably expressed genes in leukocytes obtained from healthy, physically active men to be used as solid reference genes. Genes were analyzed before and after 12 weeks of endurance training, combined with omega-3 fatty acids supplementation, to evaluate their effects on physical performance [14].

In silico analysis was performed using RefFinder, a web multi-tool that incorporates five methods: NormFinder, BestKeeper, geNorm, comparative ΔCq value, and RefFinder comprehensive ranking. Each method uses a different algorithm. NormFinder results are presented with a stability value (S), whereby a lower stability value represents relatively more stable expression [17]. The BestKeeper algorithm is based on the standard deviation (SD) and coefficient of variance and results with lower (SD) values are thought to provide better reference genes [18]. The GeNorm method is based on the exclusion of the least stable reference gene and results are presented as expression stability (M value), where lower M values indicate better stability [7]. Comparative ΔCq ranking calculates the average of standard deviation (Average STDEV), and again, lower Average STDEV values translate into more stable expression of the tested gene [19]. The most recent tool is RefFinder, which incorporates all of the aforementioned algorithms and enables a comprehensive analysis based on the geomean of ranking values (GM). A lower GM indicates greater stability [20]. We chose six potential reference genes for further analysis, based on published literature: GAPDH, ACTB, TUBB, RPS18, UBE2D2, and HPRT1. We aimed to compare commonly used reference genes with alternative genes identified in recent literature. GAPDH, ACTB, and RPS18, which are widely used in research due to their historical use in Northern Blots, were also assessed, even though their stability has been questioned by some [21]. Our results suggest that GAPDH, TUBB, and HPRT1 are the most stably expressed genes after an endurance training intervention and n-3 PUFAs supplementation period. ACTB consistently proved to be the least stably expressed. These results could enable other researchers to choose their reference genes more accurately in further studies, with regards to both physical activity and/or supplementation trials, which would make published data more accurate and easier to reproduce.

## 2. Results

### 2.1. mRNA Levels of the Candidate Reference Genes

The Cq value was obtained for six potential reference genes using the qRT-PCR method in both the omega-3 supplemented and placebo groups. The range of expression levels in the supplemented and control groups are presented in Figure 1 for all tested genes.

The observed Cq values of all tested genes ranged from 14.4 (ACTB) to 33.7 (UBE2D2). Lower Cq values suggest a notable abundance of a tested gene within the analyzed samples. GAPDH showed the lowest Cq values with a mean value of 20.07, while the HPRT gene showed the highest mean value of 25.15. The biggest range within the gene was identified for RPS18 (range = 15.14), which can be a preliminary indicator of stability, as mentioned by Giri A. and Sundar I.K [22]. No statistically significant differences between tested groups were found.

### 2.2. Evaluation of Candidate Reference Genes’ Expression after 12-Week Intervention

Since no statistically significant differences in Cq values were found between the placebo and omega-3-supplemented groups, all sample data were used for RefFinder in silico analysis. According to NormFinder, the most stable gene is GAPDH with a stability value (S) of 0.73. Other genes showed higher stability values: TUBB (S = 1.62), RPS18 (S = 1.86), UBE2D2 (S = 2.05), HPRT (S = 2.14), and the least stable ACTB (S = 5.48).

BestKeeper showed similar results when considering the SD [± crossing point values] with values presented in descending order, as follows: GAPDH (SD = 1.79), TUBB (SD = 1.84), HPRT (SD = 1.88), RPS18 (SD = 2.32), UBE2D2 (SD = 2.68), and ACTB (SD = 3.85).

The geNorm stability value (M) presented different results, identifying TUBB and HPRT as the most stably expressed genes with the same stability value of M = 0.74. GAPDH was ranked as being less stable with a result of M = 1.42, followed by RPS18 (M = 1.86), and UBE2D2 (M = 2.32). Still, geNorm also listed ACTB (M = 3.46) as the least stably expressed gene. As described above, ACTB consistently proved to be the least stably expressed gene by all integrated methods. GAPDH was the most stable according to all methods except for the geNorm algorithm.

Comparative delta-Ct ranking presented as the Average of standard deviation (Average STDEV) places GAPDH first, with the most stable result of Average STDEV = 2.64. The second most stably expressed gene was TUBB with a result of 2.83, followed by RPS18 (Average STDEV = 3.05), HPRT (Average STDEV = 3.09) and UBE2D2 (Average STDEV = 3.40). ACTB was listed at the bottom of the ranking with an Average STDEV number of 5.74. Nevertheless, RefFinder comprehensive ranking, which calculates the geomean of ranking values (GM), rated GAPDH as the best reference gene with a GM value of 1.32. The next suitable gene was TUBB (GM = 1.68), then HPRT (GM = 2.78), RPS18 (GM = 3.46), UBE2D2 (GM = 4.73), and ACTB (GM = 6.00). The results are presented in Figure 2.

## 3. Discussion

The effects of omega-3 supplementation and physical activity on human health and performance have been extensively researched over the last 10 years. Both of these factors have proven to be beneficial for human health, especially in regard to the prevention and management of civilization diseases, such as obesity, cardiovascular diseases, and mental health disorders [23,24].

Yet, few of these studies are based on long-term (>7–12 weeks), high-dose supplementation, even though both of these factors seem to be crucial to promote notable changes.

It has been previously demonstrated by Browning et al., 2012 that the amount of time needed for EPA and DHA incorporation in platelets varies between 4 and8 weeks, in the erythrocyte membrane after a minimum of 8 weeks, and in blood mononuclear cells after 6–9 months [25]. This study underscores the importance of an extended supplementation period. We have obtained similar results regarding fatty acid composition in erythrocytes. Both EPA and DHA as % of fatty acids in erythrocytes increased after a 12-week omega-3 supplementation period to a level that is considered within a target range (specific data has been shown and discussed by Tomczyk et al., 2023 [14]). This proves a physiological change and confirms the efficacy of the used dosage amount and duration in the participants of our study.

The effect of omega-3 fatty acids consumption, specifically on gene expression, is also well described in the literature; however, long-term and high-dose studies are scarce. For instance, Myhrstad et al., 2014, conducted a study in which 36 subjects ingested 8 g of either fish oil, including 1.6 g of DHA + EPA (n = 17) or sunflower oil (n = 19) for 7 days [26]. Microarray analysis was used to investigate the effect of fish oil supplementation on the transcriptome profile in PBMCs, before and after the 1-week experimental period. According to the authors, subjects were also tested after 3 weeks of supplementation. Interestingly, the authors claim that their results varied more between groups after 1 week of supplementation than after 3 weeks, which stands in opposition to data demonstrated by Browning et al., 2012 [25].

A long-term study was performed by Schmidt et al., who used qRT-PCR and microarrays to test whole-genome gene expression profiles after a 12-week exposure to high doses of n-3 PUFAs (1.14 g DHA and 1.56 g EPA) in normo- and dyslipidemic men [27]. In this study, identification of the composition of fatty acids in red blood cell membranes showed no statistically significant differences. This finding differs from the results found in the subjects of our study [14]. For qRT-PCR, Schmidt et al., [27] chose GAPDH and ribosomal protein S2 (RPS2) as reference genes based on the geNorm algorithm. The authors found increased expression of genes encoding antioxidative enzymes and a decrease in the expression of genes encoding prooxidative enzymes.

More extensive and detailed research was conducted by Bouwens et al., 2009, which involved a 26-week intervention. They tested the influence of EPA and DHA as well as high-oleic acid sunflower oil on gene expression. PBMCs from a total of 111 subjects were tested at two different doses of omega-3 fatty acids: 1.8 g EPA + DHA/d (n = 36), 0.4 g EPA + DHA/d (n = 37) and a placebo group: 4.0 g high-oleic acid sunflower oil (HOSF)/d (n = 38) [16]. A high intake of EPA + DHA was effective in altering 1040 genes, while a lower intake of fish oil influenced the expression of 298 genes. The affected genes highlight the possible anti-inflammatory and antiatherogenic properties of fish oil consumption, which is commensurate with the studies mentioned above. Moreover, this research clearly demonstrated the importance of finding stably expressed genes, since ingesting n-3 PUFAs affects the expression of a great number of genes. Normalization of the mRNA levels by suitable reference genes is a crucial step and one that affects the final reported results, as shown in many published reports [22,28,29].

In this study, we tested the stability of six potential reference genes in leukocytes obtained from healthy men, who were exposed to 12 weeks of endurance training, coupled with a high dose of omega-3 fatty acids supplementation (2.234 g EPA and 0.916 g DHA). As mentioned before, despite the abundance of literature, it is impossible to identify a single reference gene adequate for different interventions. However, we believe that the identification of possible candidate genes can, across replicate studies, allow researchers to better target reference genes in their own experimental work and may also guide interventional strategies based on the genes identified.

To the best of our knowledge, this is the first in vivo study to identify optimal reference genes in human leukocytes after a standardized endurance training and omega-3 supplementation protocol. To do this, we employed an online tool (RefFinder) that incorporates four different algorithms in order to compare the Cq values of GAPDH, ACTB, TUBB, RPS18, UBE2D2, and HPRT1.

One of the most widely used reference genes is the GAPDH gene, which encodes glyceraldehyde 3-phosphate dehydrogenase. This enzyme is involved in the process of glycolysis and in several non-metabolic processes, such as activation of transcription, initiation of apoptosis [30], or rapid axonal or axoplasmic transport [31]. Likewise, the β-ACTIN gene (ACTB) is characterized by stable expression, as it encodes a highly conserved protein that is involved in cell mobility, structure, and integrity [32]. HGPRTase, encoded by the HPRT1 gene, plays a crucial role in recovering purines from degraded DNA to reintroduce them into purine synthesis pathways [33]. The RPS18 gene carries information about the ribosomal protein, a component of the 40S subunit, and is involved in the binding of fMet-tRNA and thus, initiation of the translation process [34]. Another candidate gene is TUBB. It encodes beta tubulin protein, which is implicated in maintaining the structure of microtubules [35]. The Ubiquitin-conjugating enzyme E2 D2 is a protein that in humans is encoded by the UBE2D2 gene. Protein ubiquitination regulates the degradation of misfolded, damaged, or short-lived proteins and is mediated by a cascade of enzymes that includes E2 (ubiquitin coupling) enzyme. UBE2D2 is claimed to be one of the most stable reference genes [36].

Apart from geNorm, all of the algorithms included in the RefFinder tool showed that the single most stably expressed gene in this study was GAPDH. The MIQE guidelines suggest using more than one reference gene for more reliable results [9]. The need for using several reference genes was also discussed by Leal et al., 2015 [28].

The comprehensive data from all four software algorithms showed that GAPDH, TUBB, and HPRT are the most stable genes and using all three could be beneficial to obtain the most valid results. Moreover, GAPDH, HPRT, and TUBB are genes from different functional classes, which minimizes the risk of co-regulation. According to Vandesompele et al. [7], this diversity adds to the study’s robustness. The credibility of a single software package for choosing optimal reference genes is inconclusive and thus, RefFinder was chosen as it offers the benefits of applying and comparing multiple algorithms simultaneously. Some authors reported identical results for NormFinder, BestKeeper, and those algorithms used by RefFinder, while some show substantially different outputs for all three algorithms in and outside of RefFinder [37,38,39]. However, it should be noted that RefFinder does not take qRT-PCR efficiency data into account and De Spiegelaere et al., 2015 found that the results are similar to those that assume 100% efficiency of input data. This could possibly hinder the current findings and must be taken into consideration. Even though ACTB has been thoroughly tested as a reference gene, published data regarding its stability is inconclusive [40,41]. Our results consistently identified ACTB as the most unstable gene for the n3-PUFAs supplementation and endurance training intervention. For all that, our suggestion is that the contradictions found in the data regarding ACTB might be caused largely by the problem with the primer design, which was discussed in detail by Sun et al., 2012 [42].

As mentioned above, published data regarding the selection of appropriate reference genes is inconclusive. This might be due to the nature, specificity, and vulnerability of the PCR method [43]; hence, it is highly recommended to use more than one reference gene to obtain the most reliable results, preferably two or three. Other authors have emphasized the importance of choosing the right reference genes immediately prior to experimentation, with necessary adjustments to the cells, tissue, and methods being used [37,44]. It must also be stressed that there is no universally stable reference gene. In addition, results from our study cannot be directly applied to other studies, but they can help other researchers find the most suitable reference genes to normalize their data.

## 4. Materials and Methods

### 4.1. Ethics

The study was approved by the Bioethical Committee of Regional Medical Society in Gdansk (NKBBN/628/2019). The protocol was constructed according to the Declaration of Helsinki. All study participants were given an oral and written explanation of the study aims and written consent was obtained from each participant prior to the experiment.

### 4.2. Study Setting and Subjects

This study is part of a larger research project with details presented elsewhere [14]. Briefly, the effect of 12 weeks of endurance training with simultaneous omega-3 fatty acids supplementation was studied in healthy men. Participants received either omega-3 fatty acids or medium chain triglycerides (MCTs) in a daily dose of 2234 mg of eicosapentaenoic acid (EPA) + 916 mg of docosahexaenoic acid (DHA) (OMEGA-3 group) or 4000 mg of MCTs (PLACEBO group). Before and after intervention, blood samples were collected for omega-3 index (O3I) assessment, a sum of EPA and DHA expressed as a percent of total fatty acids in erythrocytes, which is a valid biomarker of omega-3 PUFA status. Moreover, a graded exercise test to exhaustion with assessment of VO_2_peak, running economy (RE), and a 1500-m run trial, was conducted. Out of 40 eligible participants, 26 male runners (37 ± 4 years old; 77 ± 10 kg body weight; VO_2_peak 54.2 ± 6 mL·kg^−1^·min^−1^) completed the protocol. Blood samples for gene expression analysis were collected in a fasted state, before and after the 12-week experimentation period. In the final gene analyses, 18 individuals were included (n = 10 in the OMEGA-3 group and n = 8 in the PLACEBO group). Detailed data on the inclusion and exclusion criteria are described by Tomczyk et al., 2023 [14]. The research protocol and the exclusion criteria for the gene analyses are presented in Figure 3.

### 4.3. Blood Collection, RNA Extraction and Reverse Transcription

A modification of previously described protocols for blood collection and RNA extraction were employed [45,46], with different reagents and lab equipment used in the current study. To obtain leukocytes, 2 mL of venous blood was collected from each participant into vacutainers spray-coated with K3EDTA. Within 15 min of collection, the blood was mixed with Red Blood Cell Lysis Buffer (RBCL) (A&A Biotechnology, Gdynia, Poland), incubated for 15 min, and centrifuged at 3000× *g* at 4 °C for 10 min. The obtained platelet was washed and later lysed using Fenozol (A&A Biotechnology, Gdynia, Poland). Finally, samples were stored at −20 °C for up to 3 months. Further RNA isolation was carried out using the modified Chomczynski and Sacchi method [47]. A total of 200 µL of chloroform (POCH, Gliwice, Poland) was used, samples were centrifuged, and the aqueous phase was mixed with 500 µL of isopropanol (POCH, Gliwice, Poland) and spun again. The obtained platelet was washed with ethanol, dried, and resuspended with PCR-grade water. Gel electrophoresis was performed to check for the quality and integrity of selected RNA. Nucleic Acid purity and concentration were determined. At UV 260/280, a ratio of 1.75–2.2 was accepted and at UV 260/230, a ratio >1.8 was accepted as pure RNA suitable for further analysis, based on available data [40,48]. At this stage, 4 samples from the supplemented group and 4 samples from the placebo group were excluded from further analysis due to the unacceptable UV 260/280 ratio or insufficient amount of obtained material.

Reverse transcription was performed using the AffinityScript qPCR cDNA synthesis kit (Agilent Technologies, Warszawa, Poland) and applied according to the manufacturer’s protocol with 1000 ng of RNA. The obtained cDNA was immediately frozen at −20 °C and stored for up to 1 month without repeated freeze-thaw cycles. cDNA was later diluted 1:10 with PCR-grade water immediately before the qRT-PCR step.

### 4.4. Selection of Potential Reference Genes and Primer Design

Potential reference genes (RGs) were chosen according to the data presented in published papers [3,40,41,49,50,51,52,53,54]. The genes tested in this study are listed in Table 1.

Candidate reference genes primers were either obtained from published literature or the real-time PCR primer database (PrimerBank https://pga.mgh.harvard.edu/primerbank/ (accessed on 15 November 2022)). Primer sequence, product length, and source are listed in Table 2. An efficiency of 100% was assumed for all used primers.

The specificity of potential reference genes and the obtained material was randomly checked through 2.0% agarose gel electrophoresis and later (for all samples) using a melt curve analysis.

### 4.5. Quantitative Real-Time Polymerase Chain Reaction

The AriaMx Real-Time PCR System (Agilent Technologies, Warszawa, Poland) and Brilliant III Ultra-Fast QPCR Master Mix—Agilent (Agilent Technologies, Warszawa, Poland) were used to perform qRT-PCR analyses. A total of 10 samples from the omega-3 supplemented group and 8 samples from the placebo group at 2 time points were analyzed at this stage. For the analysis, 2 µL of diluted cDNA of each sample was loaded in triplicates into 96-well PCR plates previously filled with 8 µL of MasterMix each. The thermal cycling conditions comprised an activation step: 95 °C for 10 min followed by 40 cycles of annealing, and an extension step: 95 °C for 15 s and 60 °C for 1 min. Additionally, the melt curve analysis was performed for each reaction to confirm the specific amplification of the target genes. At this point, 2 PRE-intervention samples from the supplemented group were excluded due to probable contamination (seen as Cq values > 35 and odd Tm values). On each plate, negative controls were included to verify the absence of contamination.

### 4.6. Evaluation of Stable Reference Genes for leukocytes

To compare the Cq value between the placebo and experimental groups, the D’Agostino–Pearson Normality Test was applied using GraphPad Prism version 9 for Windows, GraphPad Software, San Diego, CA, USA, www.graphpad.com (accessed on 21 January 2023). Subsequently, an unpaired t test was performed to check for statistical differences. Since no statistically significant differences were found, all Cq values were included in further analysis. In accordance with the Real-time PCR Data Markup Language (RDML), we have used the abbreviation for quantification cycle value (Cq) instead of the cycle threshold value (Ct) [58].

To evaluate the most stable reference genes, the RefFinder tool was used [20]. The RefFinder is an online software tool that integrates algorithms from the NormFinder, BestKeeper, and geNorm programs, as well as the comparative delta-Ct method.

## 5. Conclusions

Our results show that GAPDH, TUBB, and HPRT could be suitable reference genes for studies involving physical exercise and omega-3 supplementation in humans. We do not recommend using ACTB as a reference gene, based on current findings, as well as data presented in the literature. We believe there is a strong need for long-term (>7–12 weeks) molecular studies on this topic to accommodate the expected time course of adaptation. The information gained would enable a better understanding of the interplay between n-3 PUFAs supplementation and endurance training, and how these factors co-regulate changes in mRNA levels that, ultimately, mediate functional aspects of human health and performance.

## Figures and Tables

**Figure 1 ijms-24-06734-f001:**
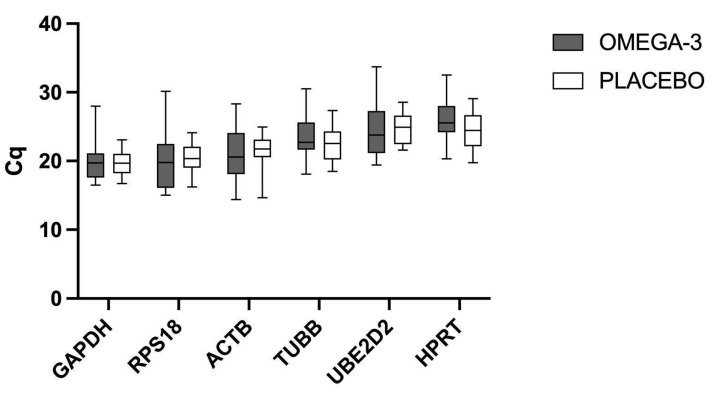
Comparison of Cq values for omega-3 supplemented vs. placebo group. The box plot for each reference gene represents the median, interquartile range, and the upper and lower range of raw Cq values for each experimental group. No statistically significant differences were found between groups.

**Figure 2 ijms-24-06734-f002:**
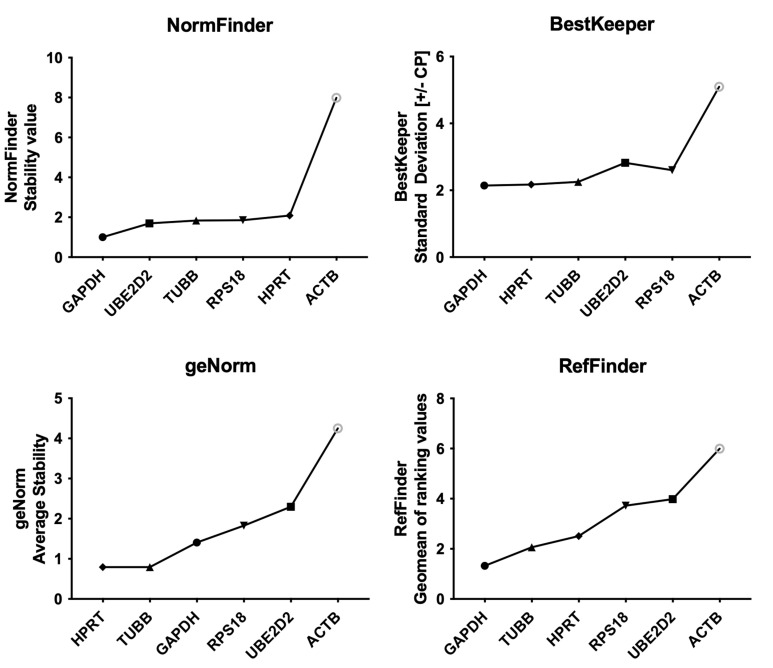
A comparison of results obtained via the RefFinder tool. The most stably expressed genes are represented by lower values obtained by three algorithms: NormFinder, BestKeeper, geNorm, and a comprehensive ranking of all algorithms together with the comparative Delta-Ct method using RefFinder.

**Figure 3 ijms-24-06734-f003:**
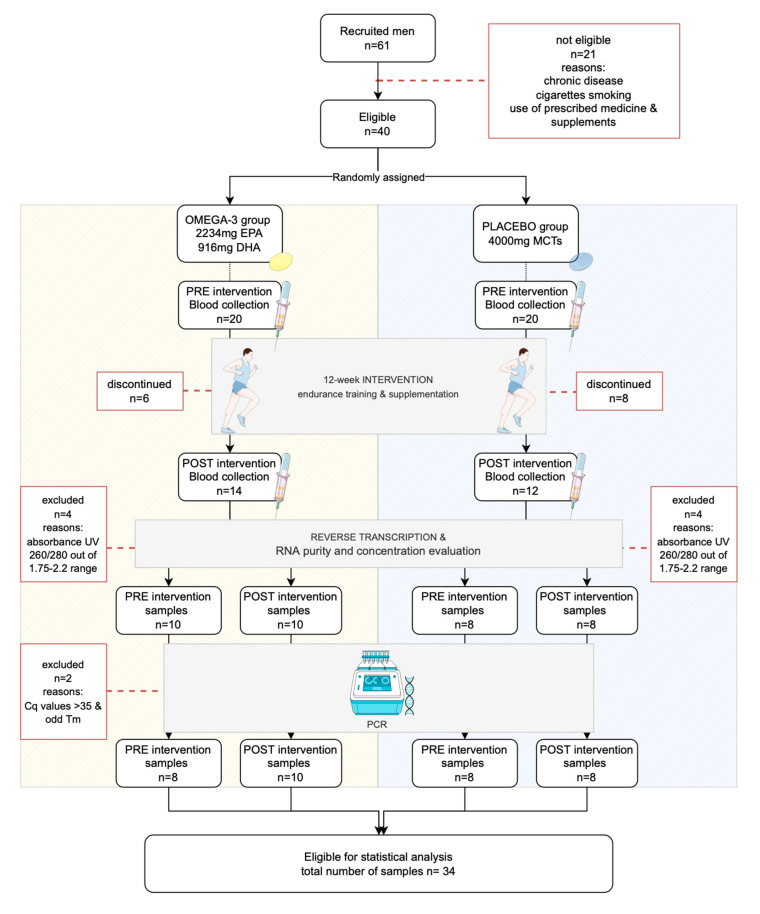
Flow diagram of the study process and group sizes.

**Table 1 ijms-24-06734-t001:** List of candidate reference genes evaluated in this study. Function information based on data published at the human genome database https://www.genecards.org/ (accessed on 15 November 2022).

Gene Symbol	Gene Accession Number	Name	Function
ACTB	NM_001101	Beta actin	Cytoskeletal protein
GAPDH	NM_002046	Glyceraldehyde-3-phosphate dehydrogenase	Oxidoreductase in glycolysis and gluconeogenesis
RPS18	NM_022551.3	Ribosomal Protein S18	Encodes a ribosomal protein that is a component of the 40S subunit
TUBB	NM_001293212.2	Tubulin Beta Class I	Forms a dimer with alpha-tubulin and acts as a structural component of microtubules
UBE2D2	NM_003339.3	Ubiquitin-conjugating enzyme E2 D2	Degradates misfolded, damaged, or short-lived proteins in eukaryotes
HPRT1	NM_000194.3	Hypoxanthine Phosphoribosyltransferase 1	Plays a central role in the generation of purine nucleotides through the purine salvage pathway

**Table 2 ijms-24-06734-t002:** Primer sequences, amplicon size and source for the sequences for each of the tested genes.

Symbol	Primer Sequence	Amplicon Size	Source
ACTB	F: GAGAAAATCTGGCACCACACC	177	Chen et al., 2018 [55]
	R: GGATAGCACAGCCTGGATAGCAA		
GAPDH	F: TCTCCTCTGACTTCAACAGCGAC	126	Andersen et al., 2004 [17]
	R: CCCTGTTGCTGTAGCCAAATTC		
RPS18	F: GCGGCGGAAAATAGCCTTTG	139	Spandidos et al., 2010 [56]
	R: GATCACACGTTCCACCTCATC		
TUBB	F: CTAGAACCTGGGACCATGGA	191	Żychowska et al., 2021 [57]
	R: TGCAGGCAGTCACAGCTCT		
UBE2D2	F: GTACTCTTGTCCATCTGTTCTCTG	120	Roy et al., 2020 [40]
	R: CCATTCCCGAGCTATTCTGTT		
HPRT1	F: CGAGATGTGATGAAGGAGATGG	97	Jeon et al., 2019 [41]
	R: TGATGTAATCCAGCAGGTCAGC		

## Data Availability

The data presented in this study are available upon request from the corresponding author.

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
