# Peer review of "Identification of Optimal Reference Genes for qRT-PCR Normalization for Physical Activity Intervention and Omega-3 Fatty Acids Supplementation in Humans"

_ijms, 2023, doi:10.3390/ijms24076734_

Round 1

Reviewer 1 Report

The manuscript is a good work on the identification of genes in human leukocytes. The manuscript is written well but the discussion section needs substantial improvement. My comments and suggestions are given in the pdf attached.

Reviewer 2 Report

Stable reference genes or housekeeping genes are essential for qRT-PCR normalization to accurately evaluate target genes or differentially expressed genes to compare different biological/experimental conditions. Robust reference genes thus ensure the accuracy and reproducibility of sequencing data in research. In this paper, authors identified optimal reference genes from leukocytes of healthy men taking omega3fatty acid supplementation. Candidate reference genes were selected from published papers, their stability was first confirmed by Cq values in lab and subsequently validated by 4 in silico tools. This reference genes can help other researchers to identify genes which are variably expressed under omega3fatty acid or other similar supplementations in humans.

Figure 1: Flow diagram clearly explains study design.

My Comments:

1.     Introduction section should answer few more questions and be rearranged

a.     What are reference genes, their importance in qRT-PCR.

b.     Fullform of MIQE

c.     What is Cq value, is this only parameter to measure qRT-PCR, its importance.

d.     Reference Gene Identification process

e.     1-2 lines description of each in silico tool parameter used for reference gene validation/gene stability indicator. For eg. geNorm: lower M value is better.

f.      And then finally the case study: stably expressed reference genes from leukocytes after omega 3 fatty acid supplementation [some information from discussion section lines 234-236 could be added here]

2.     ‘A variation of the previously described protocol for blood collection and RNA extraction have been

    applied [12,13], but different reagents and lab equipment were used in the current study.’

     I am little confused with ‘but different’ in above sentence, what does that mean?

    Using different reagents was the variation from protocol? OR

    Blood collection and RNA protocol was different and but same reagents and lab equipment used?

3.     Line 278, in discussion section mentions that ‘ReFfinder does not consider efficiency’? It seems qRT-PCR efficiency. Could you please clarify or briefly explain.

4.     Figure 2 includes Cq value of boxplots for samples after 12 weeks of intervention from both protocols.

It would be good to see the Cq values of pre-invention samples also.

5.     Line 73, if current study is part of larger research project, it would be better to provide little more information about it instead of saying details are presented elsewhere.

ENGLISH EDITING:

Sentence structure could be improved:

1.     Line 24: help other researchers choose :  ‘researchers to choose’

2.     Line 38 : is fundamental to obtaining reliable should be ‘fundamental to obtain’ or ‘fundamental for obtaining’

3.     Line 42 : Using multiple reference genes : use of multiple

4.     Provided their written consent : written consent was obtained from study participants

5.     Line 213: exposition to high doses : exposure to

MINOR COMMENTS:

1.     Try to arrange gene symbols in same order for table 1 and table 2.

2.     Order the genes based on lower to higher CQ ranges in figure 2.

3.     The Cq values mentioned in line 162-164 have ‘,’ instead of decimal point. It should be 14.4 and not 14,4.

4.     Line 177-182 results of 4 in silico tools can be summarized in tabulated format for comparative interpretation like below:

NormFinder(S)

Bestkeeper(SD)

geNorm(M)

RefFinder(GM)

ACTB

5.48

3.85

3.46

6

GAPDH

Round 2

Reviewer 2 Report

The revised manuscript addresses all my questions. 

Minor comment:

1.  Figure 2 includes Cq value of boxplots for samples after 12 weeks of intervention from both protocols.

It would be good to see the Cq values of pre-invention samples also.

Figure 2 (currently called Figure 1) presents the value of boxplots for samples before and after 12 weeks of intervention for both groups. Including both time points (pre- and post-intervention results) is a preliminary way of checking for stability as mentioned in the text below Figure 1.

Now figure 1 shows 2 plots. seems same plot just rearranged based on Cq range (as per my  minor comment). But, if they are 2 plots for  pre and post intervention, then should be labelled accordingly as 1a Pre intervention and 1b  Post intervention. if pre and post comparison needs to be made then its better to follow the same gene order in both plots.

2. Thorough spell and typo check before accepting is recommended.  e.g line 51 yetuse space missing between yet and use.

Author Response

Please find the response attached below. Thank you
